# Development of Weed Detection Method in Soybean Fields Utilizing Improved DeepLabv3+ Platform

**Helong Yu [1,2], Minghang Che [1], Han Yu [1,*] and Jian Zhang [1,2,3,*]**

1   College of Information Technology, Jilin Agricultural University, Changchun 130118, China
2   Faculty of Agronomy, Jilin Agricultural University, Changchun 130118, China
3   Department of Biology, University of British Columbia, Kelowna, BC V1V 1V7, Canada
*   Correspondence: yuhan@jlau.edu.cn (H.Y.); jian.zhang@ubc.ca (J.Z.)

**Abstract:** Accurately identifying weeds in crop fields is key to achieving selective herbicide spraying. Weed identification is made difficult by the dense distribution of weeds and crops, which makes boundary segmentation at the overlap inaccurate, and thus pixels cannot be correctly classified. To solve this problem, this study proposes a soybean field weed recognition model based on an improved DeepLabv3+ model, which uses a Swin transformer as the feature extraction backbone to enhance the model's utilization of global information relationships, fuses feature maps of different sizes in the decoding section to enhance the utilization of features of different dimensions, and adds a convolution block attention module (CBAM) after each feature fusion to enhance the model's utilization of focused information in the feature maps, resulting in a new weed recognition model, Swin-DeepLab. Using this model to identify a dataset containing a large number of densely distributed weedy soybean seedlings, the average intersection ratio reached 91.53%, the accuracy improved by 2.94% compared with that before the improvement with only a 48 ms increase in recognition time, and the accuracy was superior to those of other classical semantic segmentation models. The results showed that the Swin-DeepLab network proposed in this paper can successfully solve the problems of incorrect boundary contour recognition when weeds are densely distributed with crops and incorrect classification when recognition targets overlap, providing a direction for the further application of transformers in weed recognition.

**Keywords:** attention mechanism; improved DeepLabv3+ model; semantic segmentation; transformer; weed recognition





## 1. Introduction

Crop growth may be affected by climatic, soil and biological factors, among which weeds are one of the main biological influences. Weeds compete with crops for nutrients, sunlight, growing space and water during their growth, and may adversely affect the growth of crops if they are not removed in a timely manner [1]. Crop yield and quality are affected by pests, diseases, and weeds, with global crop yield losses approaching 30% per year [2,3]. Effectively removing weeds is critical for growing and increasing crop yield. Traditional chemical weed control involves spraying excessive amounts of herbicides indiscriminately across a given crop field, leading to the overuse of herbicides, which is both wasteful and harmful to environment and consumer health [4–6]. If it is possible to achieve the targeted and quantitative spraying of appropriate herbicides on different kinds of weeds, weed control efficiency could be improved, and herbicide use reduced. The key issue that needs to be solved to achieve selective spraying is determining how to accomplish the real-time detection and differentiation of crops and weeds [5].

To achieve the accurate detection of crops and weeds, a few problems need to be solved [7], such as the uneven density and distribution of weeds, different lighting conditions under different weather conditions resulting in reduced recognition accuracy, similar shapes and colors of crops and weeds, and the existence of shading and overlapping

leaves [8]. In recent years, machine learning has been widely used in weed identification [9]. Traditional machine-learning-based algorithms use feature descriptors to extract object features from sensory data and use machine-learning-based classifiers for classification, detection, or segmentation [10]; these mainly include supervised learning algorithms such as k-nearest neighbor algorithm and logistic regression, as well as unsupervised learning algorithms such as clustering and principal component analysis (PCA). In detecting crops and weeds, color features, texture features, location information, and multispectral features are mainly used [11]. Deng et al. [12] combined the color, shape, and texture features of weed images to solve the problem of the low accuracy of single-feature recognition of weeds in rice fields. Ashraf et al. [13] proposed two techniques for weed-density-based image classification. The first technique used texture features extracted from a grayscale cogeneration matrix (GLCM) and obtained 73% accuracy using a radial basis function (RBF) kernel in a support vector machine (SVM), while the second method outperformed the first technique using a random forest classifier with 86% accuracy. Le et al. [14] proposed a combination of local binary pattern (LBP)-based operators for extracting crop leaf texture features. The accuracy of the combined LBP algorithm was investigated, and an accuracy of 91.85% was achieved on a multiclass plant classification task. Wendel et al. [15] used vegetation separation techniques to remove the background, followed by principal component analysis (PCA) for different spectral preprocessing to extract features; finally, an SVM was used for classification, achieving an F1-score of 0.91. Traditional machine learning techniques can be used as tools for weed identification, but these techniques require significant domain expertise to construct feature extractors from raw data [16]. This increases the difficulty of model reuse between different domains, and the recognition accuracy is not as high as that of deep learning.

Deep learning is an important branch of machine learning and convolutional neural networks are the foundation of deep learning. A convolutional neural network is a complex mesh of multiple convolutional layers, pooling layers, fully connected layers and nonlinear transforms that can automatically learn from labeled data to acquire complex intrinsic features and use them to recognize unlabeled data. For image classification, target detection, and segmentation problems, DL algorithms have many more advantages than traditional machine learning methods. Due to the similarity of crops and weeds, it is difficult to extract and select features using ML methods; deep-learning-based methods have strong feature learning ability, which can effectively solve this problem [6,16]. Huang et al. [17] used a fully convolutional network (FCN) [18] semantic segmentation model with AlexNet [19], VGGNet [20], and GoogLeNet [21] as the backbone to identify weeds in rice fields, where VGGNet had the highest accuracy. They further compared this model with patch-based and pixel-based CNN structures. The results showed that the VGG-16-based FCN model had the highest classification accuracy. Fawakherji et al. [22] proposed a three-step method based on crop and weed classification. The first step was pixel-level segmentation using ResNet-101 to separate plants [23] from the soil background, the second step was the extraction of image blocks containing plants, and the third step was the classification of crops and weeds using the UNet model with VGG-16 as the backbone network; this method achieved 80% accuracy for sugar beet. Lottes et al. [24] used an FCN with an encoder–decoder structure and merged spatial information when considering image sequences, and the model was trained using RGB and NIR images. The results showed that the method greatly improved the accuracy of crop weed classification. Similarly, Li et al. [25] proposed a method to use spatial information. They proposed a new network to identify crops in the case of dense weed distribution in a field. The network used ResNet-101 as the backbone, and introduced a short link structure to achieve multiscale feature fusion to improve the segmentation of target boundaries and fine structure. The method is general, and can be applied to different crops.

Although these deep-learning-based weed recognition methods can achieve high accuracy, there is still room for improvement. Due to the short growth cycle of weeds, weeds with different growth periods may exist simultaneously in a growth area, which requires the

model to recognize multiscale targets, and there is also the problem of insufficient features due to the partial obstruction of weed features. People have started to focus on spatial information acquisition and multiscale feature extraction as the research focus [24,25]. DeepLabv3+ [26] is the first model that introduces null convolution into the field of semantic segmentation, and achieves multiscale feature extraction through null convolution with different null rates. Ramirez et al. [1] attempted to identify weeds using a DeepLabv3 model. They compared SegNet [27], U-Net, and DeepLabv3 models for the identification of crop segmentation on sugar beet farms, and found that DeepLabv3 obtained the highest accuracy with an AUC of 89% and F1-score of 81%. Wu et al. [28] designed a vision system for segmenting abnormal leaves of hydroponic lettuce using a DeepLabv3+ model with different backbone networks, and after an experimental study, ResNet-101 produced the best results, with an average intersection ratio of 83.26%.

Previous research utilized multiscale features and spatial information to improve detection accuracy, but due to the narrow perceptual field of the convolutional operation itself, the utilization of global information is still insufficient, and there is still room to improve detection accuracy. This study aims to explore the solutions to the problems of mutual occlusion caused by the dense distribution of crops and weeds in weed segmentation, the degradation of recognition accuracy caused by similar colors of crops and weeds, and the inaccuracy of kernel edge segmentation. First, some photos of the dense distribution of weeds and crops in soybean fields were collected as datasets, and data enhancement was performed to address changes in different environments. Then, this dataset was used to investigate the above issues, and a deep learning model is proposed to make improvements for the above problem.

## 2. Materials and Methods

### 2.1. Image Acquisition

The soybean weed dataset used in this experiment was collected from a soybean experimental field at Jilin Agricultural University in Changchun, Jilin Province, China, between 9:00 and 15:00 on 10 and 16 June 2021, which corresponded to the main working hours of weeding work. The device used was a Huawei mate30 cell phone, with a shooting angle perpendicular to the ground, 60 cm from the ground, a resolution of 3000 × 4000 pixels, and JPG format images.

### 2.2. Image Preprocessing

A total of 520 images of 512 × 512 pixels were collected in this experiment, and some unclear images were eliminated, resulting in 502 images. The data images mainly contain soybean crops; graminoid weeds such as *Digitaria sanguinalis* (L.) Scop and *Setaria viridis* (L.) Beauv; broadleaf weeds such as *Chenopodium glaucum* L., *Acalypha australis* L., and *Amaranthus retroflexus* L.; as well as background consisting of soil, stones, and dead plants. Therefore, the experiment divides the segmented target into soybeans, graminoid weeds, broadleaf weeds, and the remaining unsegmented target into background. The distribution of weeds and crops in the dataset used in this experiment is complex, containing many weeds and crops shading each other, which makes identification difficult. Some pictures of the dataset are shown in Figure 1.

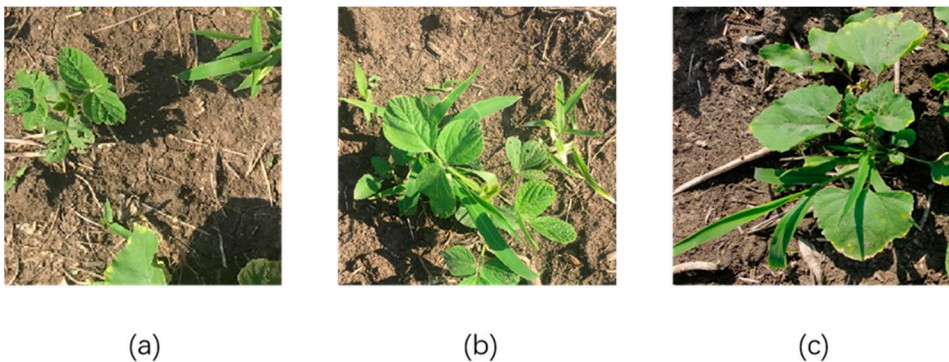

(a)                    (b)                    (c)

**Figure 1.** Examples of images of selected datasets: (**a**) image showing a sparse distribution of weeds and crops; (**b**,**c**) images showing the dense distribution of weeds and crops.

In this study, the LabelMe tool was used to label the images, and the image pixels were classified into four categories: soybeans, graminoid weeds, broadleaf weeds, and background. The labeling results are shown in Figure 2. In general, effective data expansion can better improve the robustness of a model and enable the model to obtain stronger generalization ability [29]. For this purpose, this study expanded the images using random rotation, random flipping, random cropping, adding Gaussian noise, and increasing contrast to obtain 2410 new images, which were randomly divided into a training set and a test set with a ratio of 7:3.

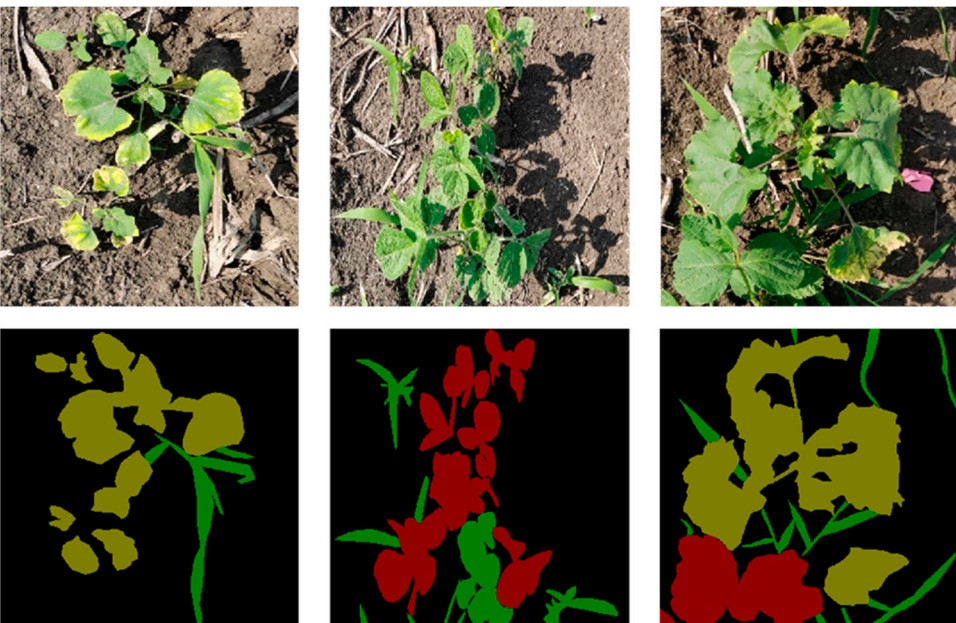

**Figure 2.** The images on the top are the original images, and the images on the bottom are their labeled images. The RGB values in the labeled images are (0,0,0) for the background area (black), (128,0,0) for soybeans (red), (0,128,0) for graminoid weeds (green), and (128,128,0) for broadleaf weeds (yellow).

### 2.3. Model Structure

Semantic segmentation has proven its effectiveness in a large number of experiments for detecting the contours of irregularly sized and shaped objects; in this study, the DeepLabv3+ model, which can fuse multiscale features, was used as the base segmentation model for segmenting soybean and weeds. DeepLabv3+ [26] is the classical semantic segmentation model, it is a more powerful encoder–decoder structure proposed by the original authors of the DeepLab series based on the DeepLabv3 [30] model as the encoder, which uses the Xception [31] model as the backbone network for feature extraction

and successively obtains a shallow feature map and a deep feature map. After obtaining the deep feature map, an atrous spatial pyramid pooling (ASPP) module with different void rate convolution operations is used to collect information at different scales in the deep feature map to enhance the recognition of features at different scales. Thus, the deep feature map processed by the ASPP module is fused with the shallow feature map to reduce the loss of edge information at the decoder stage; this could improve the accuracy of the segmentation boundary. Finally, upsampling is performed to obtain the pixel-level classification results.

The network structure proposed in this study is shown in Figure 3; it consists of two stages: encoder and decoder. The distribution of weed leaves in the image data is radial, and the leaves are far apart from each other, which will lead to the insufficient extraction of model contextual feature information. To address this issue, we used the Swin transformer to replace the backbone network of the original DeepLabv3+ model. The Swin transformer can capture the remote dependencies between pixels, utilize the spatial information, and increase the segmentation accuracy.

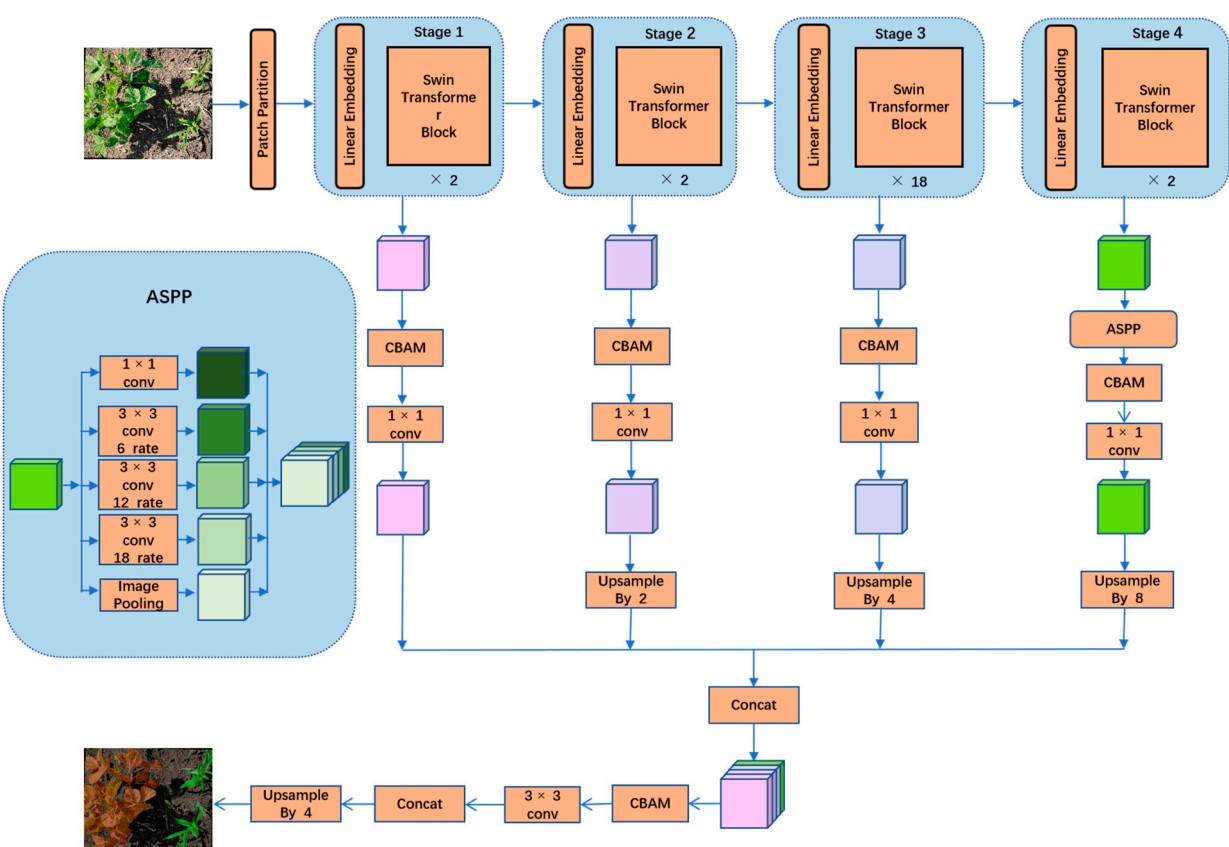

**Figure 3.** Structural diagram of the Swin-DeepLab model.

The encoder stage is divided into 4 building layers with 2, 2, 18, and 2 Swin transformer blocks, respectively. In the first layer, the size of the input feature map is $512 \times 512 \times 3$. We use the Patch Partition layer to cut the feature map in units of 4 pixels along its height (H) and width (W), and the input feature map is divided into patches with feature size of $4 \times 4 \times 3$; then, each patch is flattened along the channel direction, and the size of the input feature map becomes $128 \times 128 \times 48$. The output feature map is computed using a Linear Embedding layer to project its feature dimension to a specific dimension C (the size of C in this experiment is 96), then the Swin transformer block is used to compute the self-attentiveness of the feature map and generate the output feature map.

In the last three construction layers, to generate multi-dimensional feature maps, the feature maps are first cut into multiple patches using Patch Merging, then they are

reorganized and linearly transformed to halve the feature map dimension and double the number of channels; finally, the output feature maps are computed using the corresponding number of Swin transformer blocks. The input feature maps of the four construction layers are as follows. The feature maps generated by the four stages are a shallow feature map $\frac{H}{4} \times \frac{W}{4} \times C$, a subshallow feature map $\frac{H}{8} \times \frac{W}{8} \times 2C$, a subdepth feature map $\frac{H}{16} \times \frac{W}{16} \times 4C$, and a deep feature map $\frac{H}{32} \times \frac{W}{32} \times 8C$, and the four feature maps are used to recover the feature maps in the decoding stage.

In the decoding stage, the deep feature map is used as the input to the ASPP module. Then, the four feature maps are upsampled using the deconvolution layer. Next, the feature maps are all resized to $\frac{H}{4} \times \frac{W}{4} \times 48$ and stitched by channel dimension using the concat operation to obtain a total feature map of $\frac{H}{4} \times \frac{W}{4} \times 192$, followed by a $3 \times 3$ convolution kernel to compress their channels to $\frac{H}{4} \times \frac{W}{4} \times 48$. Finally, quadruple upsampling is used to restore the feature maps to their original size. Among them, after the four feature maps are generated and after the four feature maps are stitched together, the CBAM is added to increase the focus of the model on the channels and the key regions in space. This model is named Swin-DeepLab.

### 2.3.1. CBAM

In the weed identification task, the color features of the individual segmented objects are very similar, the shape features change due to the leaves shading each other, and the accuracy of the segmentation of the leaf edges is insufficient. To capture key information in the images, allow the algorithm to focus more on plant morphological features in space, and enhance the use of focused channel information, the model uses a CBAM [32] to deepen the attention of feature maps of different sizes generated by the backbone network. The CBAM structure is shown in Figure 4. The CBAM is a channel–space dual-attention lightweight module Similar to SENet [33], which can improve the model performance with only a small increase in computation and can be directly inserted into the existing network architecture. Therefore, we use the CBAM attention module for each dimensional feature map after its generation to refine the feature map and increase the effect of feature fusion.

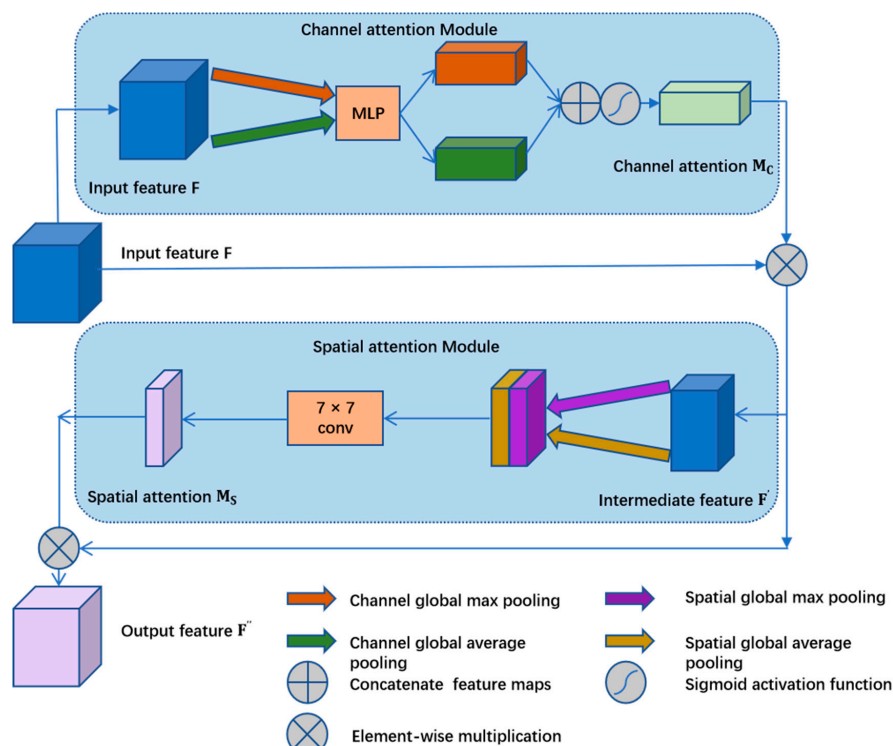

**Figure 4.** CBAM structure diagram.

The CBAM infers the attention map sequentially along two separate dimensions (channel and space), and first captures the channel features through the channel attention computation module to make the model pay more attention to the channels with more important semantic information. In the channel attention computation module, the original feature map $F^{H \times W \times C}$ is first computed by global average pooling (AvgPool) and global maximum pooling (MaxPool), separately, to generate two different feature maps $F_{avg}^c$ and $F_{max}^c$ of size $C \times 1 \times 1$, which extract channel features with their own focus. Then, the two feature maps are dimensionally compressed and recovered using a multilayer perceptron with shared weights, first compressing to the size of $\frac{C}{16} \times 1 \times 1$ and then restored to the size of $C \times 1 \times 1$. This process can increase the correlation between the channels, and finally the two feature maps are added elementwise after the activation of the sigmoid activation function to obtain the channel weight vector. The generated channel weight vector performs an elementwise summation operation with the original feature map to increase the weight of the key channel in the original feature map and generate the intermediate feature map, where the channel attention mechanism is calculated as follows:

$$M_c(F) = \sigma(\text{MLP}(\text{AvgPool}(F)) + \text{MLP}(\text{MaxPool}(F))) \tag{1}$$

where $\sigma$ denotes the sigmoid function and MLP is a fully connected layer.

Next, we use the spatial attention calculation module to calculate the spatial attention of the intermediate feature map, and the calculation process involves making the global average pooling (AvgPool) calculation and the global maximum pooling (MaxPool) calculation for the channel dimension to achieve the compression of the spatial dimension, obtain two feature maps $F_{avg}^s \in R^{H \times W \times 1}$ and $F_{max}^s \in R^{H \times W \times 1}$, then concatenate them in the channel dimension and perform convolution operations on them using a convolutional kernel of size 7 (a large convolutional kernel is used here to extract more contextual information). Finally, after activation by the sigmoid activation function, the spatial attention weight map $M^s \in R^{H \times W \times 1}$ is obtained, and the spatial attention map is used to conduct an elementwise multiplication operation with the intermediate feature map to obtain the final feature map containing more plant morphological features map $F''$, where the spatial attention mechanism is calculated by the following formula:

$$M_s(F) = \sigma\left(f^{7 \times 7}([\text{AvgPool}(F); \text{MaxPool}(F)])\right) \tag{2}$$

where $\sigma$ denotes the sigmoid function and $f^{7 \times 7}$ represents a convolution operation with a convolutional kernel size of 7.

### 2.3.2. Swin Transformer Block

A Swin transformer block is a new transformer model that includes sliding window theory, and the structure is shown in Figure 5. To reduce the computational effort and increase the computation of self-attention in locally focused regions, two consecutive Swin transformer blocks with different functions are used for each complete self-attention computation. One block is based on multihead self-attention (W-MSA) and one is based on shifted-window-based multihead self-attention (SW-MSA). The W-MSA module uses the window as the basic unit for the calculation of self-attention. This reduces the amount of calculation, but will lead to a lack of information exchange between adjacent windows; therefore, the feature map is then input into the SW-MSA module, which breaks the new window formed by the original window and performs the attention calculation within the newly formed window, thus breaking the blockage of information exchange between the original windows and enabling the exchange of information across windows.

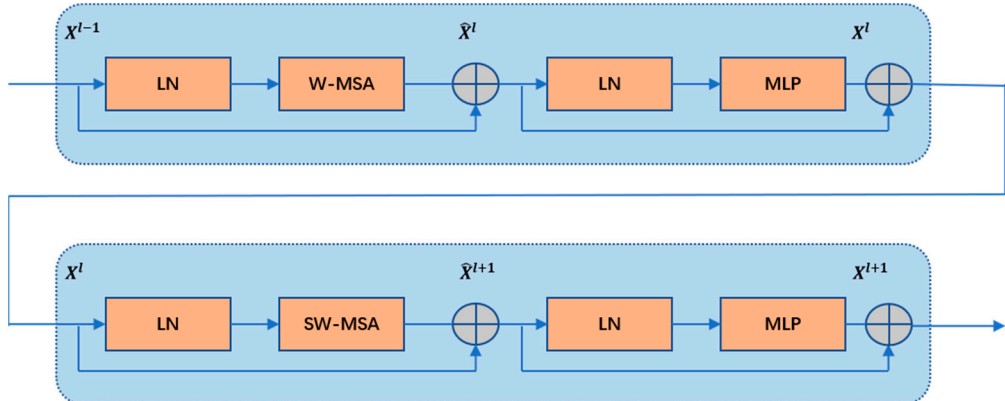

**Figure 5.** Structure diagram of the Swin transformer block.

In each Swin transformer block, the W-MSA module or SW-MSA module is followed by a multilayer perceptron, and the feature maps are normalized with a LayerNorm operation before being processed by the MSA and multilayer perceptron to prevent gradient disappearance. A jump connection is used in each Swin transformer block, and the two consecutive Swin transformer blocks are calculated as follows:

$$\hat{X}^l = W - \text{MSA}\left(\text{LN}\left(X^{l-1}\right)\right) + X^{l-1} \tag{3}$$

$$X^l = \text{MLP}\left(\text{LN}\left(\hat{X}^l\right)\right) + \hat{X}^l \tag{4}$$

$$\hat{X}^{l+1} = \text{SW} - \text{MSA}\left(\text{LN}\left(X^l\right)\right) + X^l \tag{5}$$

$$X^{l+1} = \text{MLP}\left(\text{LN}\left(\hat{X}^{l+1}\right)\right) + \hat{X}^{l+1} \tag{6}$$

$\hat{X}^l$ and $\hat{X}^{l+1}$ represent the calculation results of W-MSA and SW-MSA, $X^l$ and $X^{l+1}$ represent the calculation results of MLP, LN is the LayerNorm layer, MLP is a fully connected layer after one cycle to complete one intrawindow and interwindow attention calculation.

## 3. Experiments and Analysis of Results

### 3.1. Model Training

The hardware environment was an Intel(R) Xeon(R) Gold 6246R CPU,16 GB memory, and GPU was NVIDIA Quadro RTX 8000 GPU with an NVIDIA Quadro RTX 8000 dedicated graphics card with 48 GB of graphics memory. The software environment was Windows 10, Python version 3.8.13, PyTorch version 1.7.1, and CUDA version 11.3.

The slender leaves of grassy weeds occupy a small area in the pictures, which can lead to problems of positive and negative sample imbalance. Therefore, the cross-entropy loss function was used to calculate the loss in this experiment, and its calculation formula is as follows. To prevent the network parameters from falling into local minima during the training process, a stochastic gradient descent algorithm was used, with the initial learning rate set to 0.0001, the learning rate decay factor to 0.001, the batch size set to 16, and the number of training iterations to $100 \times 10^3$ Iter times (where one Iter is equal to one training using batch size samples).

$$Cross\_entropy = -\frac{1}{N} \sum_i \sum_{c=1}^{M} y_{ic} \log(p_{ic}) \tag{7}$$

where $M$ is the number of categories; $y_{ic}$ is the sign function, taking 1 if the true category of sample i is equal to c and 0 otherwise; and $p_{ic}$ is the predicted probability that the observed sample $i$ belongs to category $c$.

*3.2. Evaluation Indicators*

In this study, four common metrics for semantic segmentation are used to evaluate the accuracy of the model segmentation results: the mean intersection over union (mIoU), accuracy rate (Acc), precision (Pr), recall (Re), and segmentation time (ST, i.e., the time taken by the model to segment a single image in the test set). The formulae for several indicators are as follows:

$$\text{mIoU} = \frac{\sum \text{TP}}{\sum \text{TP} + \sum \text{FN} + \sum \text{FP}} \times 100\% \tag{8}$$

$$\text{Acc} = \frac{\sum \text{TP} + \sum \text{TN}}{\sum \text{TP} + \sum \text{TN} + \sum \text{FN} + \sum \text{FP}} \times 100\% \tag{9}$$

$$\text{Pr} = \frac{\sum \text{TP}}{\sum \text{TP} + \sum \text{FP}} \times 100\% \tag{10}$$

$$\text{Re} = \frac{\sum \text{TP}}{\sum \text{TP} + \sum \text{FN}} \times 100\% \tag{11}$$

TP is true positive; TN is true negative; FP is false positive; FN is false negative.

*3.3. Analysis of Results*

3.3.1. Ablation Experiments

To explore the impact of different improvement points on the performance of the model, we compared the original DeepLabv3+ model (with a ResNet50 backbone), the DeepLabv3+ model with a modified backbone (with the backbone replaced by a Swin transformer with a similar number of parameters as ResNet50), and Swin-DeepLab (which uses a Swin transformer as the backbone and adds different dimensional feature extraction and a CBAM to the decoding part) for image recognition. The results of the split are shown in Figure 6.

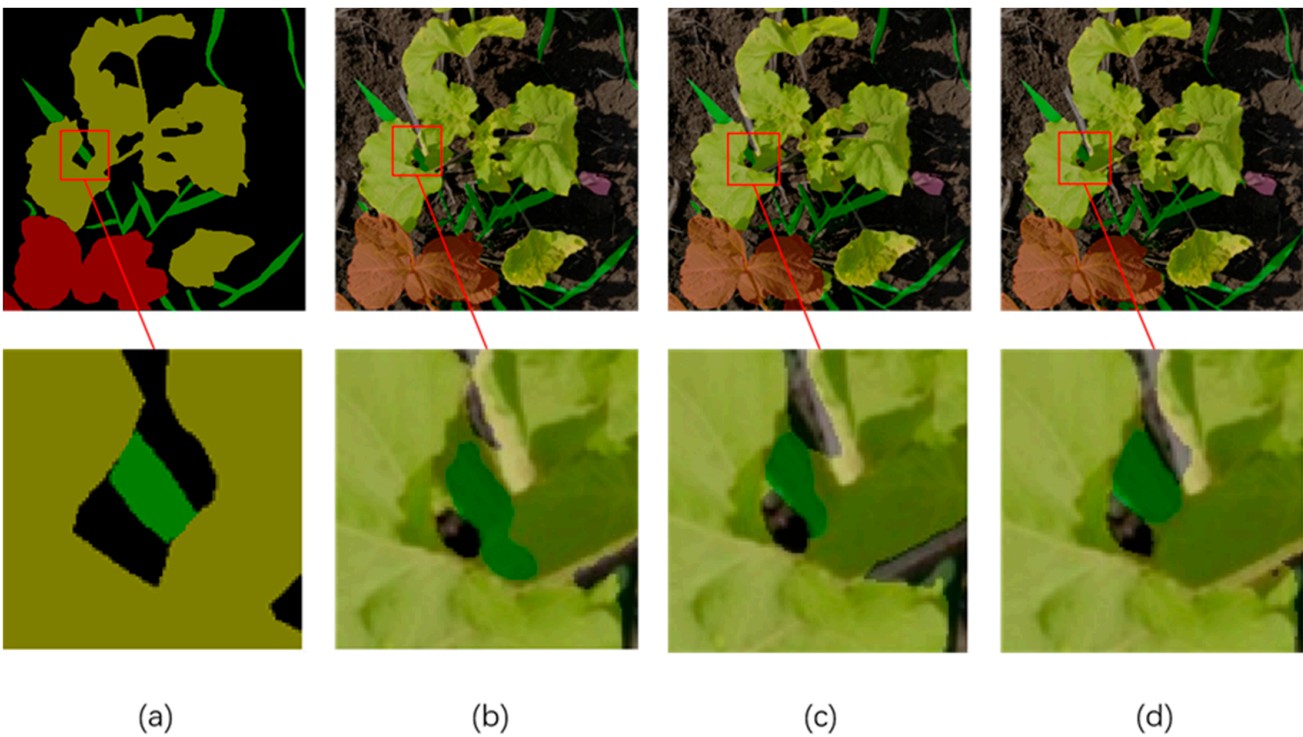

**Figure 6.** Recognition results of the model adding different improvement points: (**a**) labeled images; (**b**) DeepLabv3+ model (backbone is ResNet50); (**c**) DeepLabv3+ model with backbone modification only (backbone is Swin); (**d**) Swin-DeepLab.

In the original DeepLabv3+ model, a small number of graminoid weed pixels sandwiched between broadleaf weeds were not correctly segmented, resulting in the misclassification of neighboring broadleaf weeds. After replacing the model backbone in DeepLabv3+, the misclassification of broadleaf weeds was improved, but the graminoid weeds were still misclassified. In contrast, after adding different dimensional feature extraction and incorporating the CBAM, the contour boundaries of both weeds were better segmented, and the best results were obtained. The segmentation accuracy of the three models is shown in Table 1.

**Table 1.** Performance comparison of the models with different improvement points added.

| Model | mIoU (%) | Acc (%) | Pr (%) | Re (%) | ST (ms) |
|---|---|---|---|---|---|
| DeepLabv3+ | 88.59 | 93.40 | 94.32 | 93.10 | 319 |
| Swin+DeepLabv3+ | 91.10 | 95.15 | 95.75 | 94.33 | 341 |
| Swin-DeepLab | **91.53** | **95.48** | **95.92** | **94.58** | 367 |

### 3.3.2. Comparison of Swin-DeepLab with Other Algorithms

To verify the advantages of the Swin-DeepLab model for weed detection in soybean fields, the classical semantic segmentation models U-Net, DeepLabv3+, and PSPNet as well as the recently proposed semantic segmentation models UPerNet, CCNet, and OCRNet were selected for comparison experiments and compared with the results of the Swin-DeepLab model using the same experimental environment and data preprocessing. First, the loss curves of the models were compared, and the results are shown in Figure 7. The loss of the training set of the Swin-DeepLab model reached convergence after $50 \times 10^3$ iterations, and the loss of the test set converged relatively slowly, after $80 \times 10^3$ iterations. After reaching convergence, the loss of the validation set of the Swin-DeepLab model was the smallest. Next, the prediction result accuracy of the models was compared, as shown in Table 2. Swin-DeepLab model obtained the best results in several major metrics of semantic segmentation, reaching 91.53% for mIoU, 95.48% for Acc, 95.92% for Pr, and 94.58% for Re, surpassing the results obtained using the original model DeepLabv3+ and other comparison models. Swin-DeepLab achieved the highest segmentation accuracy at a slightly higher ST.

To investigate the effectiveness of the Swin-DeepLab model for densely distributed weed detection, the image detection results were analyzed, and the results are described below for different levels of weed distribution complexity. First, the comparative segmentation results under the weed sparse distribution condition are shown in Figure 8. In this image, the two weeds are somewhat spaced apart, not shaded by each other, and the whole plant features are more complete. In the segmentation results, the PSPNet and U-Net models still had some shortcomings in the recognition accuracy of broadleaf weeds. The Swin-DeepLab, DeepLabv3+, OCRNet and UPerNet models all basically achieved the pixel-level semantic segmentation of weeds. This figure shown that most of the evaluated segmentation algorithms could achieve good results under the conditions of sparse weed distribution.

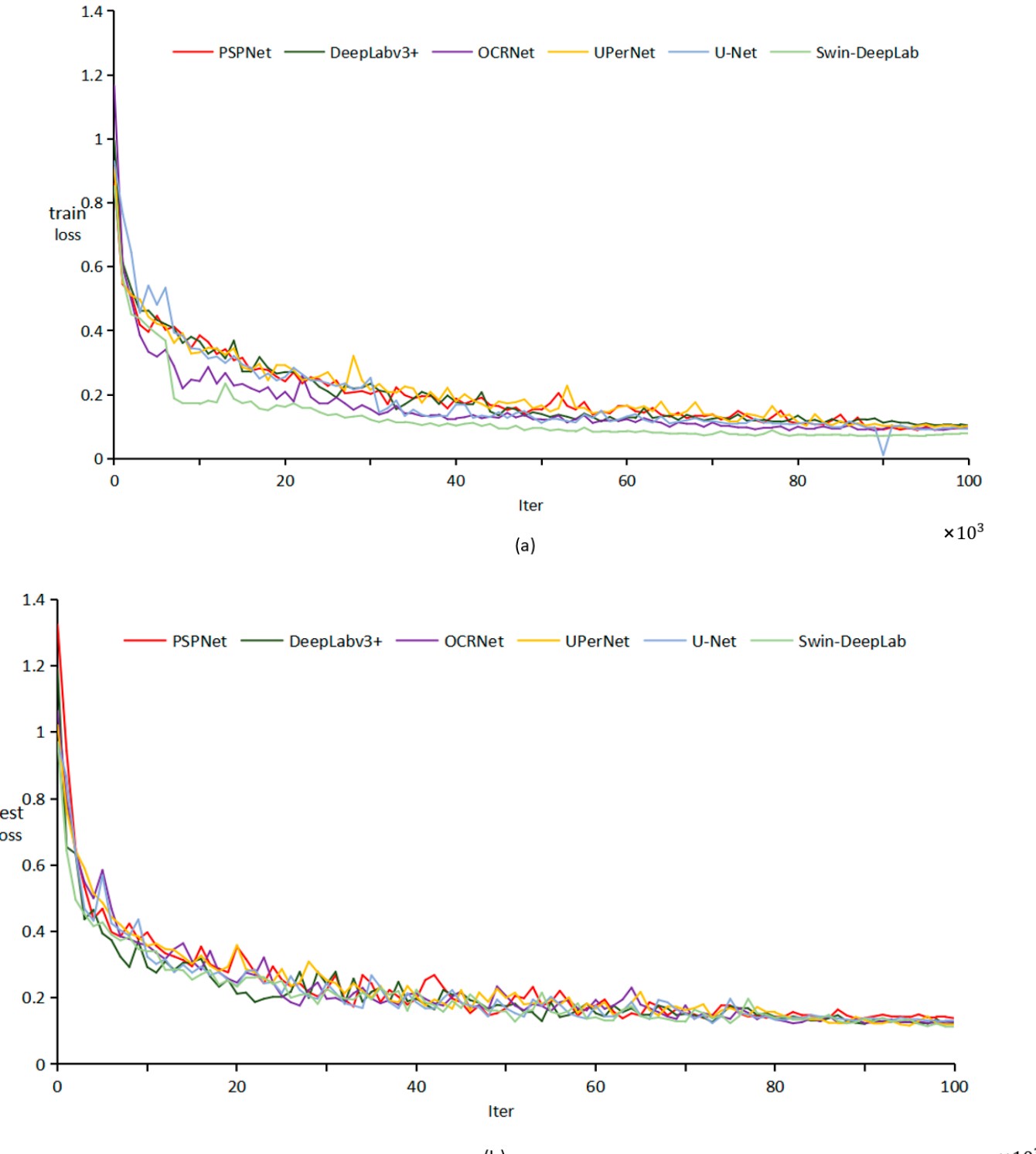

**Figure 7.** Loss variation plots: (**a**) loss plots for the training set; (**b**) loss plots for the test set.

**Table 2.** Comparison of performance metrics of the different models.

| Model | mIoU (%) | Acc (%) | Pr (%) | Re (%) | ST (ms) |
|---|---|---|---|---|---|
| U-Net | 86.61 | 92.65 | 93.58 | 92.63 | 356 |
| PSPNet | 88.18 | 93.01 | 94.11 | 93.36 | 301 |
| DeepLabv3+ | 88.59 | 93.40 | 94.32 | 93.10 | 319 |
| UPerNet | 88.75 | 92.87 | 95.14 | 92.72 | 338 |
| CCNet | 87.62 | 93.17 | 93.20 | 93.19 | 340 |
| OCRNet | 88.73 | 93.15 | 94.69 | 93.13 | 335 |
| Swin-DeepLab | **91.53** | **95.48** | **95.92** | **94.58** | 367 |

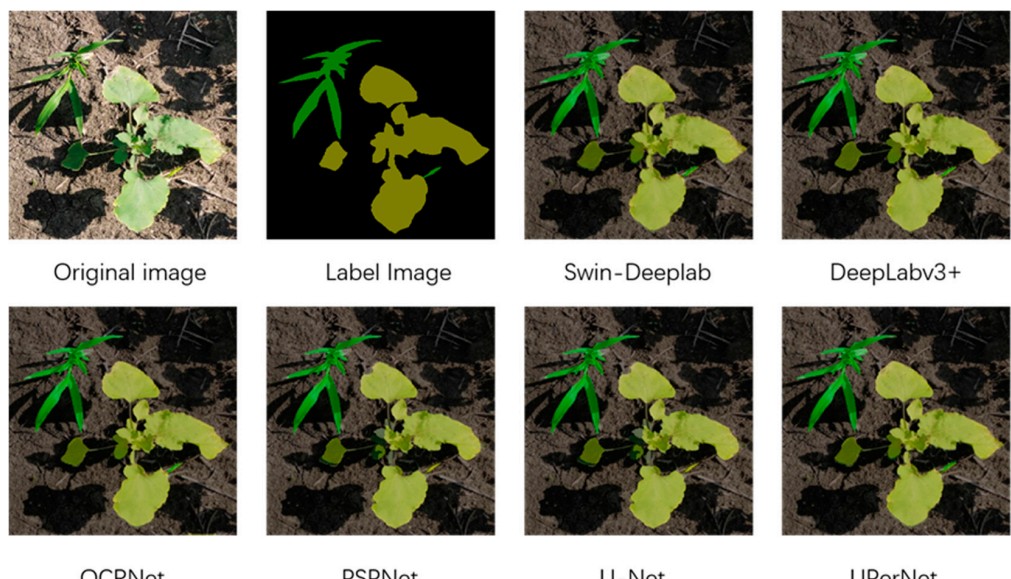

**Figure 8.** Comparison of segmentation results of different models under sparse weed distribution.

The segmentation results were then compared for the more complex spatial distribution of weeds. In this case, the soybean and weed plants were close to each other, the leaves shaded each other, the shape information of the plants was incomplete, and the segmentation accuracy decreased. The segmentation effect of different models is shown in Figure 9. PSPNet, U-Net, and DeepLabv3+ models all failed to identify weeds due to insufficient information about their leaf features. Due to the small range of spatial information extracted by the convolution operation, it is highly likely that different species of plants in proximity are regarded as the same species. In both the OCRNet model and the UPerNet model, a portion of the middle graminoid weeds was identified as the soybean leaf that surrounded it. Using the Swin-DeepLab model with the Swin transformer block as the backbone feature extraction network instead of the convolution operation, the model extracted more distant feature information, resulting in better segmentation of the overlapping portion. The use of the CBAM and the use of more intermediate feature maps for feature recovery in the decoding process resulted in more accurate leaf contours in the reduced result maps. Given the above analysis, it can be concluded that Swin-DeepLab provides better segmentation results in the case of a more complex spatial distribution of weeds.

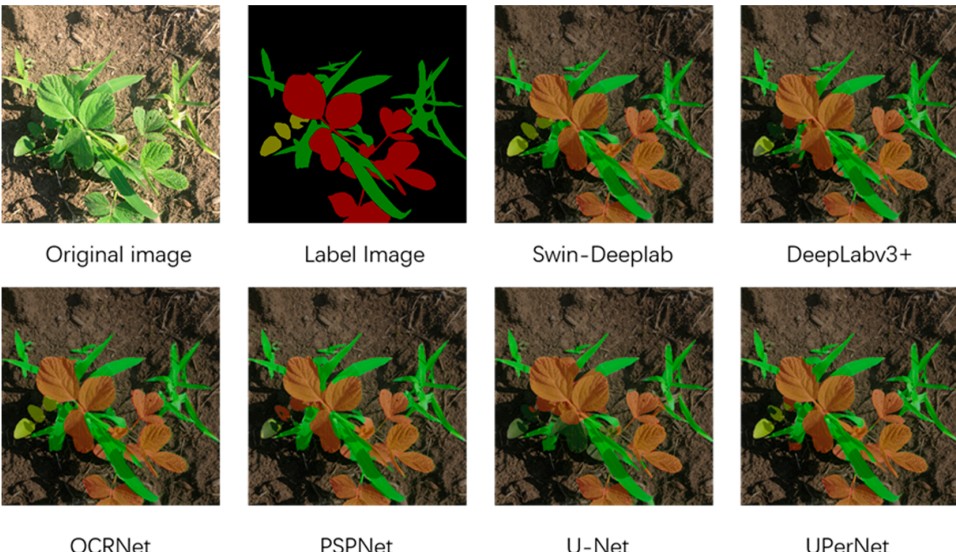

**Figure 9.** Comparison of segmentation results of different models under a dense distribution of weeds.

## 4. Discussion

In this paper, in order to solve the problem of dense distribution of weeds commonly found in actual production, we first collected a dataset of weed images from soybean fields which were collected and preprocessed to simulate different real-world conditions, then proposed a new segmentation model. Experiments were conducted using weed pictures. The model is based on the DeepLabv3+ model and replaces its original visual backbone Xception with a Swin transformer to enhance the model's key information extraction capability, which expands the perceptual field without losing the advantages of translation invariance and hierarchy of convolution operation, extracts different levels of feature maps in the decoding section, and introduces a CBAM to enhance the effect of feature recovery. This expands the perceptual field without losing the advantages of translation invariance and hierarchy of convolution operation, extracts different levels of feature maps in the decoding section, and then the CBAM attention mechanism module is used for all feature maps of different sizes so that it increases the weight of the focused semantic channels and the utilization of spatial information by the model, which effectively improves the segmentation accuracy at the junction of target contours and the segmentation accuracy in the densely distributed weed area. In the experiments, the proposed Swin-DeepLab model achieved a 91.53% mIoU and a 95.48% Acc on a complexly distributed soybean weed dataset, which is 2.94% and 2.08% better than the original DeepLabv3+ model, respectively. The new model had a detection speed of 367 ms for 512 × 512 images, which can meet the needs of practical production. This study demonstrates the capability of a vision transformer structure in weed identification, and provides direction for further applications of a vision transformer in weed identification tasks. Further research will be conducted on the application of a vision transformer in the field of crop identification, applying a vision transformer to different crop weed datasets and further improving the model to enhance the accuracy of weed segmentation, providing directions for further applications in precision agriculture.

**Author Contributions:** H.Y. (Helong Yu), M.C., H.Y. (Han Yu) and J.Z. conceived and designed the manuscript. H.Y. (Helong Yu) and M.C. analyzed the data. H.Y. (Helong Yu), H.Y. (Han Yu) and J.Z. wrote the paper. H.Y. (Han Yu) and J.Z. revised the paper. All authors have read and agreed to the published version of the manuscript.

**Funding:** This research was funded by the National Natural Science Foundation of China (grant number 42001112) and the Science and Technology Development Program of Jilin Province (grant number 20200301047RQ).

**Institutional Review Board Statement:** Not applicable.

**Informed Consent Statement:** Not applicable.

**Data Availability Statement:** Not applicable.

**Conflicts of Interest:** The authors declare no conflict of interest.

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
