# Peer review of "Development of Weed Detection Method in Soybean Fields Utilizing Improved DeepLabv3+ Platform"

_agronomy, doi:10.3390/agronomy12112889_

Round 1

Reviewer 1 Report

The article "Development of Weed Detection Method in Soybean Fields Utilizing Improved DeepLabv3+ Platform" by Yu et al. is interesting. the comparative view on different models including improved DeepLabv3+ looks promising. 

I have very few concerns as mentioned below

"Deep learning is an important branch of machine learning, and for image classification, target detection and segmentation problems, DL"  - The abbreviation DL for deep learning needs to be introduced.

  Figure legends need to be well elaborated -  Figure 5. CBAM structure diagram.   Authors have scope to further elaborate the discussion. 

Author Response

Dear reviewer,

   we would like to thank you for your valuable and constructive comments which helped us to further enhance the excellence of this research. We appreciate your time and efforts and valuable comments. We have carefully considered your comments, and made revisions according to your advice in the new version again. The details are stated as follows. We hope that the revision is acceptable and look forward to hearing from you.

Thanks in advance

Sincerely yours,

Point 1: "Deep learning is an important branch of machine learning, and for image classification, target detection and segmentation problems, DL". The abbreviation DL for deep learning needs to be introduced.

Response 1: Thank you very much for your careful review. Based on your suggestion, we have added more about DL in the third paragraph of the introduction section, , and mark it with "Track Changes".

Point 2:  Figure legends need to be well elaborated -  Figure 5. CBAM structure diagram.   Authors have scope to further elaborate the discussion. 

Response 2: Thank you very much for your useful suggestions, based on your suggestion, we have extensively revised the introductory section of the CBAM module to provide a more detailed elaboration of the entire computational process,  moreover  the discussion section explains more about the role of CBAM and mark it with "Track Changes".

Reviewer 2 Report

Line number

Comment

27

Arrange keywords into alphabetical order

32

Better to mention as excessive weeds are one of the key factors……; because there may be other major factors for controlling the yield.

33

Better to mention as resources; because, weeds compete for light, space also, not only for nutrients and water.

63

What is mean by PCA? Define them when you first introduce 

70

Explain what is mean by DL; if it mean deep learning, better to indicate that in 69 line.

72

What is mean by ML?

113

No need to explain method of experiment in introduction.

128

Explaining about achievements in this section is not good. Better to write it as objectives or mention achievements in conclusion section.

151

Better mention the scientific names of those weeds

201

Does 512x512x3 figure compress into 128x128x48? how 3 come into 48?

215

What is mean by H and W?

246

Better to explain symbols here.

335

ST means?

354

What is mean by val loss (y axes in figure b)?

-          If it possible explain materials and method section in more simple way.

-          Maintain same font even in figure descriptions as well; in 227, 250, 286, 323, 352, 353, 384 and 385.

Author Response

Dear reviewer,

   we would like to thank you for your valuable and constructive comments which helped us to further enhance the excellence of this research. We appreciate your time and efforts and valuable comments. We have carefully considered your comments, and accordingly made revisions in the new version again. The details are stated as follows. We hope that the revision is acceptable and look forward to hearing from you.

Thanks in advance

Sincerely yours,

Corresponding authors on behalf of all authors

Point 1: Arrange keywords into alphabetical order

Response 1: Thank you very much for your careful review. With your suggestion, we have reordered the keywords alphabetically, and mark it with "Track Changes".

Point 2: Better to mention as excessive weeds are one of the key factors……; because there may be other major factors for controlling the yield.

Response 2: We are grateful for your excellent advice. With your suggestion, we have modified the introduction of the main factors affecting the growth stages of the crop and mark it with "Track Changes".

Point 3: Better to mention as resources; because, weeds compete for light, space also, not only for nutrients and water.

Response 3: We are grateful for your good advice. Following your suggestion, w added light and spatial factors to the description of the effect of weeds on crops and mark it with "Track Changes".

Point 4: What is mean by PCA? Define them when you first introduce 

Response 4: Thank you very much for your careful review. Based on your suggestion, we use the full name rather than the abbreviation when referring to PCA, and mark it with "Track Changes".

Point 5: Explain what is mean by DL; if it mean deep learning, better to indicate that in 69 line.

Response 5: Thank you very much for your useful suggestions. Following your advice, we have added the definition of deep learning in the paragraph you specified.

Point 6: What is mean by ML?

Response 6: Thank you very much for your useful suggestions. To address the problem of inadequate machine learning definitions, we have added the definition of machine learning in the second paragraph, and mark it with "Track Changes".

Point 7: No need to explain method of experiment in introduction.

Response 7: Thank you very much for your useful suggestions, based on your suggestion, we have removed the part from introduction section.

Point 8: Explaining about achievements in this section is not good. Better to write it as objectives or mention achievements in conclusion section.

Response 8: Thank you very much for your useful suggestions, based on your suggestion, we have integrated it the achievement section.

Point 9: Better mention the scientific names of those weeds

Response 9: Thank you very much for your careful review. We have changed the name of these weeds to the scientific name based on your suggestion and mark it with "Track Changes".

Point 10: Does 512x512x3 figure compress into 128x128x48? how 3 come into 48?

Point 11: What is mean by H and W?

Point 15: If it possible explain materials and method section in more simple way.

Response 10.11.15: Thank you very much for your useful suggestions, we believe that suggestions 10, 11 and 15 were all due to the lack of clarity in the model introduction section, and we have extensively reworked the model introduction section based on your valuable suggestions. We have modified the third paragraph section to give the meaning of H and W according to suggestion 10, and modified it to give a clearer description of the compression process of the input feature map according to suggestion 11.

Point 12:  Better to explain symbols here.

Response 12: Thank you very much for your useful suggestions, based on your suggestion ,We added the meaning of the  LN and MLP symbols , and mark it with "Track Changes".

Point 13: ST means?

Response 13: Thank you very much for your careful review. Based on your suggestion, we added the meaning of ST, and mark it with "Track Changes".

Point 14:  What is mean by val loss (y axes in figure b)?

Response 14: Thank you very much for your careful review., This was a mistake on our part, we modify val loss to test loss , and mark it with "Track Changes".

Point 16:  Maintain same font even in figure descriptions as well; in 227, 250, 286, 323, 352, 353, 384 and 385.

Response 16: Thank you very much for your useful suggestions, based on your suggestion, we have readjusted the font of the figure in line 210, 234, 271, 308, 338, 339, 369, 371. However, due to the different sizes of images and the number of modules contained in them, resizing the fonts in all images to the same size may cause some images to look odd, so the resized fonts in different images are not exactly equal.

Reviewer 3 Report

The manuscript is well-written and explains the research clearly. Results presented are interesting and detail a topical problem, which with the improvement of weed recognition software in agricultural crops, will make farmers lives easier.

Author Response

Dear reviewer,

   we would like to thank you for your valuable and constructive comments which helped us to further enhance the excellence of this research. We appreciate your time and efforts and approval.  Your approval is very important to our work, which will motivate us to further advance this kind of study, and we will try our best.

Thanks you again.  

Sincerely yours,

Helon and Jian on behalf of all authors 

Round 2

Reviewer 2 Report

Moderate English language editing is required